# Potential of Bacterial Strains Isolated from Coastal Water for Wastewater Treatment and as Aqua-Feed Additives

**DOI:** 10.3390/microorganisms9122441

**Published:** 2021-11-26

**Authors:** Kyochan Kim, Joo-Young Jung, Jong-Hee Kwon

**Affiliations:** 1Department of Civil and Environmental Engineering, Korea Advanced Institute of Science and Technology (KAIST), Daejeon 34141, Korea; chan1028@kaist.ac.kr; 2Division of Applied Life Sciences (BK21), Gyeongsang National University, Jinju 52828, Korea; 3Department of Food Science & Technology, and Institute of Agriculture & Life Science, Gyeongsang National University, Jinju 660-701, Korea

**Keywords:** *Pseudoalteromonas mariniglutinosa*, *Psychrobacter celer*, *Bacillus albus*, *Bacillus safensis*, wastewater treatment, aqua-feed additives

## Abstract

Bacteria have various and sustained effects on humans in various fields: molecular biology, biomedical science, environmental/food industry, etc. This study was conducted to evaluate the wastewater treatment capacity and feed-additive fish-growth effect of four strains of bacteria: *Pseudoalteromonas mariniglutinosa*, *Psychrobacter celer*, *Bacillus albus*, and *Bacillus safensis*. In a wastewater degradation experiment, (i) nitrate-N and nitrite-N were removed within 1 h in all of the 4 bacterial strains; (ii) the removal rates of TAN and TN were higher in all of the strains relative to the *B. subtilis*. In a feed-additive experiment (5% Kg^−1^), (i) the growth of fish was higher in all of the 4 bacterial strains with the *B. subtilis* relative to the commercial feed; (ii) there was no significant growth difference for *B. albus* and *B. safensis* relative to the *B. subtilis*, but growth was higher in *P. mariniglutinosa* and *P. celer*. The results indicated that the 4 bacterial strains can be effectively utilized for biological wastewater treatment processes and as aqua-feed.

## 1. Introduction

Bacteria, the first living organisms on Earth, have various and sustained effects on humans in various fields. In the fields of molecular biology and biomedical science, *Escherichia coli* sp. (gram-negative) plays a role in supplying the body with menaquinone (Vitamin K); with *Mycobacterium phlei*, *Lactobacillus subtilis Natto*, and *Staphylococcus aureus*, which animals cannot synthesize [1,2,3], it produces insulin to help diabetic patients [4,5,6]; and in the environmental field, it is used as an important water-quality indicator [7,8,9]. In the chemical industry, *Bacillus* sp. (gram-positive) is utilized as an intermediary for butanediol production (e.g., *Bacillus velezensis*, *Bacillus toyonensis*, *Bacillus sapensis*, *Bacillus licheniformis*) [10,11,12]; in the environmental industry, for biological wastewater treatment (e.g., *Bacillus subtilis*, *Bacillus thuringiensis*, *Bacillus mycobacterium*) [13,14,15], and in the food industry, for fermentation and a food additive (e.g., *Bacillus licheniformis*, *Bacillus natto*, *Bacillus subtilis*, *Bacillus amyloliquefaciens*) [10,16,17].

In the wastewater treatment, a major problem to be solved is the removal of high organic loads, especially nitrogen and phosphorus, which contribute to eutrophication and, in this regard, biological treatment has received significant attention relative to physical and chemical methods, due to its environmental-friendliness, cost-effectiveness, and sustainability [18,19,20]. *Sphingomonas* sp. (gram-negative) has shown a total nitrogen (TN) removal efficiency of 94.22% in nitrification and 90.10% in denitrification processes after 48 h [21]; *Pseudomonas* sp. *mendocina* (gram-negative) removed up to 100 mg L^−1^ ammonia in 12 h with no nitrite-N (NO_2_-N) accumulation, and 100 mg L^−1^ nitrite-N (NO_3_-N) and 72.61 mg L^−1^ nitrate-N (NO_3_-N) under aerobic conditions [22]; *Acinetobacter* sp. (gram-negative) indicated excellent removal abilities for ammonium (98.5%), nitrate-N (NO_3_-N, 91.1%), TN (97.9%), and chemical oxygen demand (COD, 93.5%) under acidic conditions [23]. Other strains, namely Alcaligenes faecalis, *Rhodococcus* sp., *Thiosphaera pantotropha*, and *Alcaligenes faecalis*, also have been reported to remove nitrogen sources through biological nitrogen fixation [24,25,26,27]. In the food industry, feed additives such as probiotics, prebiotics, acidifiers and plant- or animal-derived extracts are defined as non-nutritive ingredients or non-nutritive components of ingredients [28,29,30], and the common probiotics used in aquaculture are derived from *Saccharomyces*, *Lactobacillus*, *Bacillus*, *Lactococcus*, *Shewanella*, *Carnobacterium*, *Clostridium*, *Enterococcus*, *Leuconostoc*, *Aeromonas*, and several other strains [31,32,33,34]. *Bacillus* sp., *Lactobacillus* sp. for cobia (*Rachycentron canadum*), *Clostridium* sp. on white shrimp (Litopenaeus vannamei), and Shewanella sp. on abalone (*Haliotis discus hannai*) have been used as dietary probiotics to enhance growth performance, intestine antioxidant capacity, activities in serum enzymes, cellular and humoral immune response, alternative complement pathway (ACP) activity, phagocytosis percentage (PP), and respiratory burst activity [31,35,36].

This study was conducted to discover useful heterotrophic microorganisms not yet reported in many studies and, more specifically, to evaluate (i) their effectiveness in biological-wastewater-treatment and (ii) aqua-feed additive applications in terms of extents of fish growth, respectively. We evaluated (i) the removal of total ammonia nitrogen (TAN), nitrite-N (NO_2_-N), nitrate-N (NO_3_-N), total nitrogen (TN), and total phosphorus (TP) for the biological water treatment and (ii) the effect on fish growth using the microorganisms’ freeze-dried biomass obtained from mass cultivation as a feed additive. For those purposes, 4 dominant bacterial strains, namely *Pseudoalteromonas mariniglutinosa* (gram-negative), *Psychrobacter celer* (gram-negative), *Bacillus albus* (gram-positive), and *Bacillus safensis* (gram-positive) were identified.

## 2. Materials and Methods

In general, the analyses were performed in triplicate at the minimum. *Bacillus subtilis* was used as a control in the water-quality-degradation experiment, and as a semi-control in the feed-additive experiment. 

### 2.1. Site Description and Strains Collection

Microorganism samples were collected from three coastal sites at Chungcheongnamdo province (36°55′28.8′′ N, 126°48′08.0′′ E), Jeollabukdo province (35°52′07.3′′ N 126°30′29.8′′ E), and Jeju-island (33°18′10.5′′ N, 126°48′15.5′′ E) in the Republic of Korea, and five dominant (fast-growing) candidate strains cultured in LB (Luria-Bertani, LB Broth, Miller, BD Difco^TM^, Sparks, MD, USA) solid medium finally were selected: *Pseudoalteromonas mariniglutinosa* (gram-negative), *Psychrobacter celer* (gram-negative), *Bacillus albus*, *Bacillus safensis*, and *Bacillus subtilis* (Figure 1). All of the bacterial strains were confirmed by nucleotide-sequence analysis of the 16S ribosomal RNA (rRNA). Direct cycle sequencing of the purified PCR products was performed using the Big Dye Terminator v3.1 Cycle Sequencing Kit (Applied Biosystems, Foster City, CA, USA) with the primers, and a BLAST similarity search of the 16S rRNA was conducted of the National Centre for Biotechnology Information (NCBI) database using BLASTN (https://blast.ncbi.nlm.nih.gov/Blast.cgi, accessed on 12 March 2019) [37]. Analysis of the nucleotide sequences and phylogenetic tree construction was followed by construction of a Maximum Likelihood (ML) phylogenetic tree from the NCBI database library.

### 2.2. Cultivation and Harvesting of Bacteria

The five strains were cultivated in three sets of 10 L, sealed-cylindrical plastic bottles per strain with medium (glucose 5 g L^−1^, Yeast extract 3 g L^−1^, NaCl 10 g L^−1^, Tryptone 8 g L^−1^). Constant aeration (250 mL min^−1^) was provided by air stones placed on the bottom to maintain the oxygen level, and the water temperature was maintained at 28 °C. After 30 h, the bacteria were harvested by a dynamic filtration module entailing rotation of a perforated disk (FMX B-class [bench-scale, BKT Co. Ltd., Daejeon, Korea]) [38]. Upon completion of the filtration, the concentrates of bacteria were freeze-dried for 7 days in a pre-weighed tube.

### 2.3. Evaluation of Biological Wastewater Treatment

Domestic-sewage wastewater was collected from the National Municipal Wastewater Treatment Plant located in Gunsan city (35°58′14.2′′ N 126°34′22.4′′ E), and solid particles therein were removed by GF/C glass-fiber filter paper (0.45 μm, Whatman, Maidstone, UK). For the experiment, the average concentration of the five strains for the inoculum in the filtered wastewater (1 L cylindrical bioreactor) was 7.14 × 10^7^ colony forming units (CFU), and it was maintained with a constant air flow (200 mL min^−1^) at 28 °C. To induce bacterial growth, 1 g L^−1^ of glucose was added to the filtered wastewater. Samples were taken every hour to measure the total ammonia nitrogen (TAN), nitrite-N, nitrate-N, total nitrogen (TN), and total phosphorus (TP).

The supernatant was passed through a 0.2 μm syringe filter, and the TAN, nitrite-N, nitrate-N concentrations were quantified by ion chromatography (883 Basic IC Plus, Metrohm, Herisau, Switzerland, LOD 0.02~0.27 μg L^−1^), and TN and TP concentrations were measured by water-analysis kits (Nitrogen VARIO, total LR and Phosphate-total LR, Aqualytic^®^, Dortmund, Germany, > ppm). For analysis of the nitrate-N and nitrite-N, an anion column Metrosep A Supp 5 (150 mm × 4.0 mm, Metrohm, Switzerland) was used, with eluent consisting of 3.2 mM Na_2_CO_3_ and 1 mM NaHCO_3_ supplied to the column at a flow rate of 0.7 mL min^−1^. For quantification of the TAN concentration, a cation column MetrosepC4 (150 mm × 4.0 mm, Metrohm, Switzerland) with eluent containing 1.7 mM HNO_3_ and 0.7 mM PDCA (2,6-Pyridinedicarboxylic acid) supplied at a flow rate of 0.9 mL min^−1^ was used [39].

### 2.4. Preparation of Experimental Diets (5% Bacterial Feed Additives), and Fish and Rearing Conditions

Five experimental diets of freeze-dried bacteria biomass (5% Kg^−1^) added to commercial feed containing 47.44% crude protein and 5.65% crude lipid were prepared. The diets were well mixed with a blender, pelletized by a pellet-extruder, dried over the course of 72 h at 25 °C, to an approximately 9–10% moisture level, and finally stored at −20 °C until use. The experiment was conducted for 8 weeks. Thirty (30) juvenile amur catfish (*Silurus asotus*) (initial body weight, 34.2 g) obtained from a local fish farm were distributed into 200 L plastic tanks (water volume: 150 L). Experimental diets (5% Kg^−1^ of each bacteria) were supplied to the fish four times daily (9:00, 13:00, 17:00, 21:00) by satiation. The water was 40% replaced daily, constant aeration (5 L min^−1^) was provided by air stones placed on the bottom to maintain the oxygen level (DO, >5 mg L^−1^), and the water temperature (25–26 °C) and pH (7–8) were maintained under a 16-h light/8-h dark photoperiod cycle.

### 2.5. Growth Performance and Proximate Composition Analysis

At the end of the feeding trial, 10 fish were randomly selected from each tank and weighed to determine the weight gain (WG), feed efficiency (FE), specific growth rate (SGR), and survival rate. A proximate composition analysis of the samples (whole body) was performed using the standard AOAC methods [40]. Preparatorily, strains of bacteria and fish were freeze-dried for 48 h. The moisture contents were determined by means of a dry oven at 105 °C, and the ash contents by combustion at 550 °C. The crude protein was analyzed by the Kjedahl method, and the crude lipid was analyzed by soxhlet extraction using the soxhlet system 1046 (Tecator AB, Hoganas, Sweden) [41].

### 2.6. Amino Acid Compositions

Strains of bacteria and fish (whole body) were freeze-dried for amino acid (AA) analysis [42]. A total of 0.02 g of sample was hydrolyzed with 15 mL of 6 N HCl at 110 °C for 24 h. The hydrolyzed samples, in distilled water within a 50 mL flask, were evaporated and recovered in sodium citrate buffer (0.2 N, pH 2.2). After filtration (0.2 µm), the samples were analyzed with ninhydrin at 570 nm and 440 nm using a S433 amino acid analyzer (Sykam, Gilching, Germany, >ppm). For methionine and cystine hydrolysis, performic acid was used in place of 6 N HCl.

### 2.7. Fatty Acid Methyl Ester (FAME) Compositions

Freeze-dried strains of bacteria and fish powder (whole body) were treated with chloroform/methanol (2:1, *v*/*v*), and lipids were determined using the modified Folch method [41]. Methanol and sulfuric acid were added, after which the solution was incubated at 100 °C for 20 min for fatty acid methyl ester (FAME) conversion. Heptadecanoic acid was used as an internal standard. The organic phase was separated by 0.3 M NaOH, then recovered by centrifugation at 4000 rpm for 10 min. The FAMEs were measured by gas chromatography (HP 6890, Agilent, Santa Clara, CA, USA, LOD 3.9~28.7 mg L^−1^) with a flame-ionized detector and HP-INNOWax capillary column (30 m × 0.32 mm × 0.5 μm, Agilent, Santa Clara, CA, USA). The temperatures of the injection port and detector were 250 and 280 °C, respectively. The GC column temperature profile was as follows: (1) initial temperature 50 °C for 1 min; (2) increased to 200 °C at 15 °C min^−1^, held for 9 min; and (3) increased to 250 °C at 2 °C min^−1^; held for 2 min. The fatty acids were identified by comparing the retention times with those of standard fatty acids (37 Component FAME Mix, Supelco, Bellefonte, PA, USA) and quantified using the peak areas on the chromatogram with the internal standard [43]. The FAME content was calculated as
$$FAME\;content\;\left(\%,\;\frac{w}{w}\right)=\frac{weight\;of\;FAME\;obtained\;after\;transeseterification}{weight\;of\;fish\;whole\;body}\times 100$$

### 2.8. Statistical Analysis

All of the data were analyzed by one-way ANOVA (SPSS Inc., Chicago, IL, USA) to test the effects of the treatments. When a significant treatment effect was observed, a Tukey HSD test was applied for comparison of the means. The treatment effects were considered at the 5% level of significance (*p* < 0.05).

## 3. Results and Discussion

### 3.1. Identification of Strains

Hundreds of microbial strains were classified from the samples collected from the water of Chungcheongnamdo provine, Jeollabukdo province, and Jeju-island. The results of the analysis are shown in three clusters (Figure 1). After BLASTn analysis, similarity for each strain was 99% (*Pseudoalteromonas mariniglutinosa* to KMM3635, Figure 1a), 99% *(Psychrobacter celer* to SW-238, Figure 1b). 98% (*Bacillus albus* to MCCC 1A02146, Figure 1c), 100% (*Bacillus safensis* to FO-36b, Figure 1d), and 99% (*Bacillus subtilis* to JCM 1465, Figure 1e).

### 3.2. Effects of 4 Candidate Strains on Nutrient Removal

*Bacillus subtillis* is one of the commonly used bacteria in biological wastewater treatment processes [44,45,46]. Total ammonia nitrogen (TAN, ionized ammonia as nitrogen plus ionized ammonia as nitrogen, Figure 2a) was removed faster in all of the experimental groups (four strains of bacteria) relative to the control (*B. subtilis*), and the removal rate within 1 h was 97.5% (control, *B. subtilis*) and 100% (all of the experimental groups, i.e., the four strains of bacteria). Nitrite-N (NO_2_-N, Figure 2b) and nitrate-N (NO_3_-N, Figure 2c) were almost completely removed within 1 h in both the control (*B. subtilis*) and the 4 experimental groups (strains of bacteria), and the removal rate of total nitrogen (TN, Figure 2d) was in the order of *Pseudoalteromonas mariniglutinosa* (96.1%, gram−), *B. safensis* (79.9%, gram+), *B. albus* (79.9%, gram+), *Psychrobacter celer* (78.0%, gram−), and *B. subtilis* (72.6%, gram+). Considering that TN (precisely: total dissolved nitrogen, TDN) is the total amount of nitrogen compounds composed of dissolved organic nitrogen (DON) (e.g., free amino acids, urea, nucleic acids, trihalomethanes [THMs], dihaloacetic acids [DHAAs] [47,48]), dissolved inorganic nitrogen [DIN], TAN (NH_4_^+^ + NH_3_), nitrite-N (NO_2_-N), nitrate-N (NO_3_-N), or nitrous oxide [N_2_O] (depending on the oxidation state in water), and assuming, based on this, that DON is simply determined by subtracting DIN concentrations from TN [49], the order of the four strains of bacteria (four strains + control strain) for the amount and rate of dissolved organic nitrogen (DON) in wastewater degraded by them, was: *Pseudoalteromonas mariniglutinosa* (27.45 mg L^−1^, 92.3%), *B. safensis* (17.95 mg L^−1^, 60.3%), *B. albus* (17.95 mg L^−1^, 60.3%), *Psychrobacter celer* (16.85 mg L^−1^, 56.6%), and *B. subtilis* (13.65 mg L^−1^, 45.9%), respectively. The amount and rate of total phosphorus (TP, Figure 2e) removal were high in the order of *B. safensis* (5.79 mg L^−1^, 82.7%), *B. albus* (5.70 mg L^−1^, 81.4%), *B. subtilis* (5.64 mg L^−1^, 80.6%), *Psychrobacter celer* (5.62 mg L^−1^, 80.3%), and *Pseudoalteromonas mariniglutinosa* (5.53 mg L^−1^, 79.0%), and there was no significant difference between the control (*B. subtilis*) and the 4 experimental groups (strains of bacteria).

Recently, simultaneous heterotrophic nitrification and aerobic denitrification (SND) bacteria including *Bacillus sp.* have emerged as a promising approach to nitrogen removal by microorganisms in wastewater treatment [22,50,51,52]. This type of bacteria consequently shows the capability for removal of ammonium-nitrogen, nitrite-nitrogen and nitrate-nitrogen through heterotrophic nitrification (NH_4_^+^ → NH_2_OH → NO_2_^−^ → NO_3_^−^) and aerobic denitrification (NO_3_^−^ → NO_2_^−^ → N_2_O → N_2_) under aerobic conditions [53,54]. Therefore, the four strains of bacteria belong to SND bacteria, according to the present results. In addition, there are microorganisms such as ammonia-oxidizing bacteria (AOB), nitrite-oxidizing bacteria (NOB), and denitrification bacteria that prefer specific nitrogen sources for their growth, but there are also cases where they selectively absorb nitrogen depending on the environment. For example, *Pseudomonas* sp. (gram−) prefer nitrate-N (NO_3_-N)*, Bacillus* sp. (gram+) and *Gordonia* sp. (gram+) prefer ammonia, and *Chlorella* sp. and *Scenedesmus* sp. (eukaryotes, autotrophs) prefer nitrite-N (NO_2_-N) [55,56,57]. In the light of the amounts of the different nitrogen sources and the removal rates in the results of the current study (Figure 2), it was considered that all of the strains prefer TAN to the other nitrogen sources. For more accurate and comprehensive information on this, additional and more specific studies are required.

### 3.3. Evaluation of Four Strains of Bacteria as Feed Additives for Fish-Growth Performance

The growth performance and feed efficiency data on catfish fed with commercial feed and freeze-dried bacteria biomass-supplemented diets (5% Kg^−1^) are provided in Table 1. Similarly to the results of many previous studies [58,59,60], the growth of the fish fed the semi-control (*B. subtilis*-supplemented diet) was greater than that of the fish fed the control (commercial feed), suggesting that useful microorganisms, from a nutritional point of view, can have a positive effect on fish growth by playing a role as feed additives or probiotics. The overall values of weight gain (WG), feed efficiency ratio (FE), and specific growth rate (SGR) were significantly different between the *B. subtilis*-supplemented and 4 bacteria-strain-supplemented diets. This supports the contention that the amount and the type or properties of proteins affecting the digestibility of diets can have an effect on the growth of fish [28,61]. In this respect, note the following values (relative to the commercial-feed control): *B. subtilis* (29.10 g Kg^−1^), *B. albus* (21.02 g Kg^−1^), *B. safensis* (28.40 g Kg^−1^), *Pseudoalteromonas mariniglutinosa* (35.30 g Kg^−1^), and *Psychrobacter celer* (34.12 g Kg^−1^) (Table 2a).

The total protein and lipid contents in the whole catfish body ranged from 14.08 to 16.35 and from 5.23 to 6.82%, respectively (Table 2b). The protein/lipid content ranges for each group were 15.45/5.23% (*B. subtilis*), 15.41/6.26% (*B. albus*), 14.10/5.37% (*B. safensis*), 14.08/5.57% (*Pseudoalteromonas mariniglutinosa*), and 16.35/6.82% (*Psychrobacter celer*), values which were within the ranges of other, relevant studies [62,63,64,65,66].

The amino acid compositions of the microorganisms and fish fed with the commercial and microbial-supplemented diets are provided below (Table 3). As for the amino acid compositions of the microorganisms (Table 3a), all of the strains were found to contain the highest amounts of leucine (*B. subtilis*), lysine (*B. albus* and *Pseudoalteromonas mariniglutinosa)*, arginine (*B. safensis*), and histidine (*Psychrobacter celer*) with the least amounts of methionine in essential amino acids (EAA); and, all of the strains were found to contain the highest amounts of glutamic acid with the least amounts of cysteine in non-essential amino acids (NEAA). For the fish fed with the commercial and microbial-supplemented diets (Table 3b), the highest and least amounts were methionine and histidine in the EAA, and glutamic acid and cysteine in the NEAA, in all of the groups.

The fatty acid compositions of the microorganisms and fish fed with the commercial and microbial-supplemented diets are provided below (Table 4). All of the strains were found to contain the highest amounts of palmitic acid (C16:0) in both all of the strains (Table 4a) and in the fish (Table 4b). In the microorganisms (Table 4a), the saturated fatty acid (SFA) content was found to be high in the order of *P. celer* (83.37%), *B. safensis* (81.79%), *B. albus* (71.37%), *B. subtilis* (61.64%), and *P. mariniglutinosa* (61.42%); the monounsaturated fatty acid (MUFA) content, in the order of *B. subtilis* (21.39%), *B. albus* (18.13%), *P. ma`riniglutinosa* (17.21%), *B. safensis* (14.29%), and *P. celer* (11.65%), and in the polyunsaturated fatty acid (PUFA) content, in the order of *P. mariniglutinosa* (21.37%), *B. subtilis* (16.97%), *B. albus* (10.50%), *P. celer* (4.98%), and *B. safensis* (3.92%). Interestingly, arachinodic acid (C20:0, ARA, one of the components of breast milk and a precursor to inflammation-inducer prostaglandin and leukotriene) was found in *P. mariniglutinosa*; meanwhile, eicosapentaenoic acid (C20:5n3, EPA) and docosahexaenoic acid (C22:6n3, DHA), conditionally essential fatty acids with ARA for humans and animals, were found in *P. mariniglutinosa* and *P. celer*, respectively. In the fish (Table 4b), there was no significant difference between SFA and MUFA among the control (commercial feed), semi-control (*B. subtilis*), or any of the experimental groups (four strains), except for PUFA. Myristic acid (C14:0), α-Linolenic acid (C18:3n3), γ-Linolenic acid (C18:3n6), arachidic acid (C20:0), eicosatrienoic acid (C20:3n3), eicosapentaenoic acid (C20:5n3), erucic acid (C22:1n9), docosadienoic acid (C22:2), docosahexaenoic acid (C22:6n3), and lingnoceric acid (C24:0), which were not detected in microorganisms but were detected in fish, were considered to have come from the commercial feed. Overall, the bacteria-supplemented diets affected changes in the amino acid (Table 3) and fatty acid (Table 4) compositions in the fish.

### 3.4. Practical Applications and Future Research Prospects

In summation, in the environmental field, first, general wastewater treatment at a wastewater treatment plant (WWTP) is largely divided into three stages: (i) primary treatment (removal of suspended solids); (ii) secondary treatment (removal of floating or soluble organic matter by use of microorganisms), and (iii) tertiary or advanced treatment (removal of nitrogen and phosphorus that cause eutrophication). There are various physical, chemical, and biological factors such as DO (Dissolved Oxygen), BOD (Biochemical Oxygen Demand), COD (Chemical Oxygen Demand), TOC (Total Organic Carbon), NOD (Nitrogenous Oxygen Demand), solids (e.g., TS, TDS, TVS, VSS, VDS, TFS, TSS), acidity, and hardness, all of which are used as water-quality measurement parameters. In the case of the biological wastewater treatment process, the removal of TN and TP is prioritized through the nitrification and denitrification processes by microorganisms. Therefore, the microorganisms used in this study can be utilized in environmental applications, because they have been shown to be excellent for water-quality-improvement purposes.

Second, in the aquaculture field, nitrite-N (NO_2_-N) and ammonia (NH_3_) are important water-quality factors, as they incur mass mortality. Nitrite-N (NO_2_-N) occurs naturally in fresh water as a result of nitrification of ammonia and denitrification of nitrate, and its concentration is enhanced by partial oxidation of ammoniacal discharge. Its effect is manifested by the conversion of haemoglobin to methaemoglobin, which is incapable of oxygen transport; thus, nitrite is toxic to fish, including vertebrates [55,67]. As for total ammonia nitrogen (TAN), it represents the sum of un-ionized (NH_3_) and ionized (NH_4_^+^) ammonia. Acute ammonia (NH_3_) toxicity, for example, affects the central nervous system of vertebrates, leading quickly to convulsions and death [68]. Ammonia (NH_3_)’s toxicity is ascribed to its ready diffusion across gill membranes due to its lipid solubility and lack of charge, whereas ionized ammonia (NH_4_^+^) acts in a larger hydrated form with charged entities that cannot readily pass through the hydrophobic micropores in the gill membrane [69]. The toxicity of the forms of ammonia (NH_3_) increases with water pH (the ammonia/ammonium ratio increases at pH 7 and above), which is one of the reasons for the importance of pH control in aquaculture. Aquaculture systems including autotroph biofloc technology (ABFT) [19], biofloc technology (BFT) [70], recirculating aquaculture systems (RAS) [71], and integrated multi-trophic aquaculture (IMTA) [72] systems such as aquaponic systems [73], constructed wetlands [74], integrated marine systems [75], high-rate algal ponds [76], periphyton systems [77], and partitioned aquaculture systems [78] aim mainly to remove nitrite-N (NO_2_-N) and ammonia (NH_3_). Therefore, the four strains used in this study can be utilized in aquaculture, as they were shown to be excellent in their water-quality-improvement utilities.

Third, as a result of using the four strains as feed additives, growth was similar or higher than that of the control species (*Bacillus subtilis*); thus, they can be used as independent feed additives in the form of biomass (powder) in aquaculture.

Fourth and finally, the four strains of microorganisms contain essential amino acids, essential fatty acids, and bioactive substances (EPA, DHA, ARA) in their cells, and as such, can be mass-produced and used as health supplements in the food industry.

## 4. Conclusions

This study was conducted to evaluate the wastewater treatment capacity and feed-additive fish-growth effect of four strains of bacteria: *Pseudoalteromonas mariniglutinosa, Psychrobacter celer*, *Bacillus albus,* and *Bacillus safensis*. In the results of the wastewater degradation experiment, nitrate-N (NO_3_-N) and nitrite-N (NO_2_-N) were almost completely removed within 1 h in all of the experimental groups, and the removal rates of total ammonia nitrogen (TAN) and total nitrogen (TN) were higher in all of the experimental groups relative to the control (*B. subtilis*). Overall, *Pseudoalteromonas mariniglutinosa* was superior to the others. As for the feed additive (5% Kg^−1^ bacteria biomass-supplemented) experiment: relative to the control (commercial feed), the growth of fish was higher in all of the experimental groups (four strains), and, relative to the semi-control (*B. subtilis*), there was no significant difference in *B. albus* or *B. safensis*, but there was, in fact, a significant difference in *Pseudoalteromonas mariniglutinosa* and *Psychrobacter celer*.

## Figures and Tables

**Figure 1 microorganisms-09-02441-f001:**
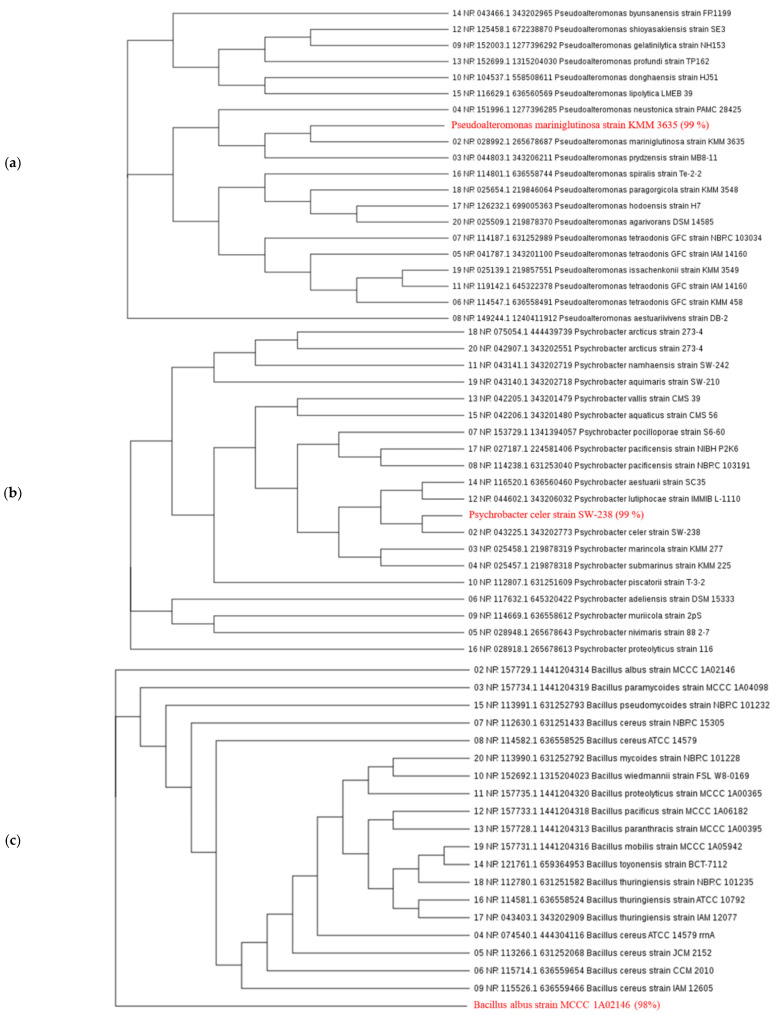
The phylogenetic tree derived from neighbor-joining analysis of partial 16S rRNA gene sequence: (**a**) *Pseudoalteromonas mariniglutinosa*, (**b**) *Psychrobacter celer*, (**c**) *Bacillus albus*, (**d**) *Bacillus safensis*, and (**e**) *Bacillus subtilis*.

**Figure 2 microorganisms-09-02441-f002:**
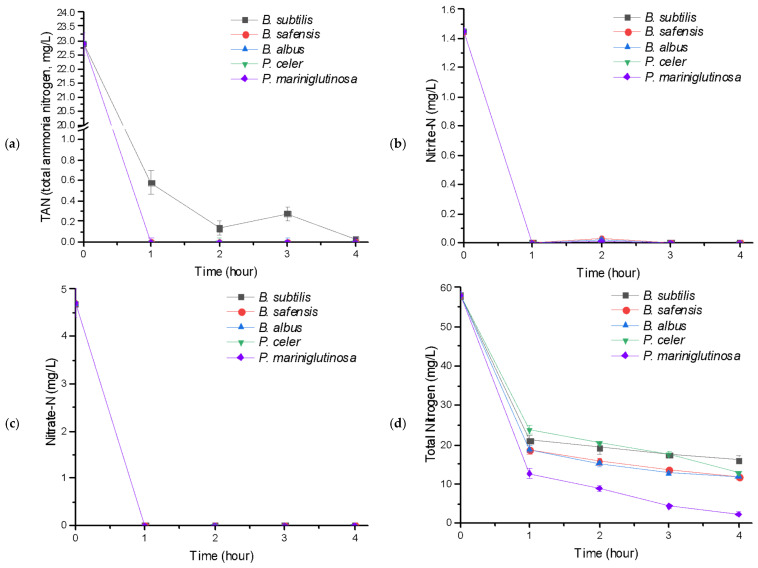
Nutrient-removal efficacy: (**a**) TAN (total ammonia nitrogen), (**b**) Nitrite-N (NO_2_-N), (**c**) Nitrate-N (NO_3_^−^-N), (**d**) Total nitrogen (TN), and (**e**) Total phosphorus (TP). Groups: *Bacillus subtilis*, *Bacillus safensis*, *Bacillus albus*, *Psychrobacter celer*, and *Pseudoalteromonas mariniglutinosa*.

**Table 1 microorganisms-09-02441-t001:** Growth performance of Amur catfish fed with commercial feed and 5% freeze-dried bacteria biomass-supplemented feed at end of 8-week experiment ^1^.

	C. F ^6^ (Con)	*B.subtilis* (Semi-control)	*B. albus*	*B. safensis*	*P. mariniglutinosa*	*P. celer*
Initial mean weight (g fish^−1^)	34.2 ± 0.1	34.1 ± 0.1	34.0 ± 0.1	34.1 ± 0.1	34.2 ± 0.1	34.1 ± 0.1
Final mean weight (g fish^−1^)	75.3 ± 1.5 ^c^	79.8 ± 1.1 ^b^	81.7 ± 1.7 ^b^	81.1 ± 1.6 ^b^	92.8 ± 2.5 ^a^	91.3 ± 2.5 ^a^
WG (%) ^2^	120.2 ± 4.5 ^d^	134.0 ± 3.3 ^c^	140.3 ± 5.0 ^b^	137.8 ± 4.6 ^bc^	171. ± 7.2 ^a^	167.7 ± 7.3 ^a^
FE (%) ^3^	123.3 ± 2.1 ^c^	137.1 ± 5.4 ^b^	143.1 ± 3.8 ^b^	141.0 ± 6.6 ^b^	175.8 ± 5.3 ^a^	171.6 ± 7.1 ^a^
SGR (%/day) ^4^	1.41 ± 0.04 ^d^	1.52 ± 0.03 ^c^	1.57 ± 0.04 ^b^	1.55 ± 0.03 ^bc^	1.78 ± 0.05 ^a^	1.76 ± 0.05 ^a^
Survival (%) ^5^	100	100	100	100	100	100

^1^ Values are means from triplicate groups of fish where the values in each row with different superscripts are significantly different (*p* < 0.05). ^2^ Weight gain (WG, %) = (final weight−initial weight) × 100/initial weight. ^3^ Feed efficiency ratios (FE, %) = (wet weight gain / dry feed intake) × 100. ^4^ Specific growth rates (SGR, %/day) = (log_e_ final weight−log_e_ initial weight) × 100/days. ^5^ Survival rate (%) = (initial number of fish−dead fish) × 100/initial number of fish. ^6^ Commercial feed containing 47.44% protein and 5.65% lipid, Korea.

**Table 2 microorganisms-09-02441-t002:** Whole-body proximate compositions of bacterial strains and Amur catfish fed with commercial feed and 5% freeze-dried bacteria biomass-supplemented feed at end of 8-week experiment ^1^.

	Microorganisms (a) ^2^					Amur Catfish (b) ^3^					
	*B. subtilis*	*B. albus*	*B. safensis*	*P. mariniglutinosa*	*P. celer*	C.F ^4^ (Con)	*B. subtilis* (Semi-control)	*B. albus*	*B. safensis*	*P. mariniglutinosa*	*P. celer*
Moisture	1.26 ^a^	1.39 ^a^	1.28 ^a^	0.16 ^b^	0.11 ^b^	76.03 ^b^	75.94 ^b^	75.76 ^b^	77.84 ^a^	78.52 ^a^	75.01 ^b^
Protein	58.36 ^c^	42.05 ^e^	56.81 ^d^	70.61 ^a^	68.34 ^b^	15.57 ^b^	15.45 ^b^	15.41 ^b^	14.10 ^c^	14.08 ^c^	16.35 ^a^
Lipid	0.56 ^a^	0.44 ^b^	0.48 ^ab^	0.43 ^b^	0.33 ^c^	6.25 ^b^	5.23 ^c^	6.26 ^b^	5.37 ^c^	5.57 ^c^	6.82 ^a^

^1^ Values are means from triplicate groups of shrimps where the values in each row with different superscripts are significantly different (*p* < 0.05). ^2^ Dry matter basis. ^3^ Wet weight basis. ^4^ Commercial feed containing 47.44% protein and 5.65% lipid, Korea.

**Table 3 microorganisms-09-02441-t003:** Amino acid compositions of bacterial strains and Amur catfish fed with commercial feed and 5% freeze-dried bacteria biomass-supplemented feed at end of 8-week experiment (%) ^1,2^.

	Microorganisms (a)					Amur Catfish (b)					
	*B. subtilis*	*B. albus*	*B. safensis*	*P. mariniglutinosa*	*P. celer*	C.F ^3^	*B. subtilis*	*B. albus*	*B. safensis*	*P. mariniglutinosa*	*P. celer*
Essential amino acids (EAA)											
Arginine	2.50 ^cd^	2.29 ^d^	3.43 ^a^	2.80 ^b^	2.52 ^c^	0.98 ^bc^	0.97 ^bcd^	1.07 ^a^	0.89 ^d^	0.91 ^cd^	1.02 ^ab^
Threonine	2.05 ^b^	1.53 ^c^	2.02 ^b^	2.25 ^a^	2.02 ^b^	0.69 ^a^	0.65 ^ab^	0.66 ^ab^	0.61 ^bc^	0.56 ^c^	0.68 ^a^
Valine	2.96 ^c^	2.13 ^e^	2.56 ^d^	3.87 ^a^	3.54 ^b^	0.73 ^a^	0.71 ^a^	0.71 ^a^	0.64 ^b^	0.57 ^c^	0.73 ^a^
Isoleucine	2.30 ^c^	1.84 ^d^	2.21 ^c^	2.92 ^a^	2.64 ^b^	0.66 ^a^	0.66 ^a^	0.66 ^a^	0.58 ^b^	0.53 ^b^	0.66 ^a^
Leucine	3.44 ^bc^	2.55 ^d^	3.19 ^c^	3.76 ^a^	3.47 ^b^	1.15 ^a^	1.11 ^a^	1.14 ^a^	1.01 ^b^	0.93 ^b^	1.19 ^a^
Methionine	0.65 ^c^	0.80 ^b^	1.19 ^a^	0.67 ^c^	0.46 ^d^	2.57 ^a^	1.86 ^b^	1.36 ^c^	1.39 ^c^	1.84 ^b^	1.42 ^c^
Lysine	3.35 ^b^	2.63 ^c^	3.13 ^b^	3.91 ^a^	3.81 ^a^	1.26 ^a^	1.25 ^a^	1.25 ^a^	1.10 ^b^	1.01 ^b^	1.27 ^a^
Phenylalanine	2.15 ^b^	1.47 ^d^	1.90 ^c^	2.33 ^a^	2.12 ^b^	0.65 ^a^	0.63 ^a^	0.64 ^a^	0.56 ^b^	0.53 ^b^	0.66 ^a^
Histidine	1.91 ^d^	2.73 ^c^	2.64 ^c^	3.30 ^b^	4.44 ^a^	0.50 ^b^	0.51 ^ab^	0.49 ^b^	0.44 ^c^	0.41 ^c^	0.54 ^a^
Total	21.31 ^b^	17.97 ^c^	22.27 ^b^	25.81 ^a^	25.02 ^a^	9.19 ^a^	8.35 ^b^	7.98 ^b^	7.22 ^c^	7.29 ^c^	8.17 ^b^
Non-essential amino acids (NEAA)											
Serine	1.87 ^a^	1.17 ^d^	1.65 ^b^	1.65 ^b^	1.47 ^c^	0.71 ^ab^	0.66 ^bcd^	0.68 ^abc^	0.63 ^cd^	0.61 ^d^	0.73 ^a^
Glutamic acid	7.24 ^c^	6.16 ^d^	7.53 ^c^	9.63 ^a^	8.48 ^b^	2.16 ^ab^	2.12 ^bc^	2.16 ^ab^	1.96 ^cd^	1.86 ^d^	2.31 ^a^
Proline	2.15 ^b^	1.47 ^c^	1.52 ^c^	2.42 ^a^	2.50 ^a^	0.87 ^b^	0.78 ^c^	0.86 ^b^	0.83 ^bc^	0.97 ^a^	0.98 ^a^
Glycine	2.40 ^b^	1.58 ^d^	2.15 ^c^	2.62 ^a^	2.41 ^b^	1.17 ^b^	1.16 ^b^	1.06 ^c^	1.16 ^b^	1.34 ^a^	1.20 ^b^
Alanine	4.17 ^b^	2.96 ^c^	3.29 ^c^	5.50 ^a^	5.17 ^a^	1.04 ^a^	1.00 ^ab^	0.94 ^b^	0.94 ^b^	0.97 ^ab^	1.03 ^a^
Tyrosine	1.24 ^b^	1.18 ^b^	1.44 ^a^	1.38 ^a^	1.19 ^b^	0.52 ^ab^	0.48 ^bc^	0.52 ^ab^	0.45 ^cd^	0.41 ^d^	0.53 ^a^
Aspartic acid	4.68 ^a^	3.57 ^c^	5.00 ^a^	4.15 ^b^	4.02 ^b^	1.50 ^a^	1.48 ^a^	1.48 ^a^	1.33 ^b^	1.25 ^b^	1.55 ^a^
Cysteine	0.71 ^b^	0.57 ^c^	0.80 ^a^	0.40 ^e^	0.49 ^d^	0.13	0.10	0.10	0.09	0.09	0.09
Total	24.46 ^bc^	18.66 ^d^	23.38 ^c^	27.75 ^a^	25.73 ^b^	8.10 ^ab^	7.78 ^abc^	7.80 ^abc^	7.39 ^c^	7.50 ^bc^	8.42 ^a^

^1^ Values are means from triplicate groups of shrimps where the values in each row with different superscripts are significantly different (*p* < 0.05). ^2^ Dry matter basis. ^3^ Commercial feed containing 47.44% protein and 5.65% lipid, Republic of Korea.

**Table 4 microorganisms-09-02441-t004:** Fatty acid compositions of bacterial strains and Amur catfish fed with commercial feed and 5% freeze-dried bacteria biomass-supplemented feed at end of 8-week experiment (%) ^1,2^.

	Microorganisms (a)					Amur Catfish (b)					
	*B. subtilis*	*B. albus*	*B. safensis*	*P. mariniglutinosa*	*P. celer*	C.F ^6^	*B. subtilis*	*B. albus*	*B. safensis*	*P. mariniglutinosa*	*P. celer*
C12:0	2.65 ^b^	3.44 ^a^	1.27 ^d^	0.34 ^e^	1.85 ^c^	0.18 ^a^	0.13 ^b^	0.13 ^b^	0.11 ^b^	0.10 ^b^	0.11 ^b^
C14:0	4.26 ^e^	5.05 ^d^	11.14 ^b^	5.94 ^c^	13.60 ^a^	5.04 ^a^	4.74 ^ab^	4.38 ^bc^	4.00 ^d^	4.01 ^d^	4.22 ^cd^
C14:1						0.06	0.06	0.07	0.06	0.05	0.06
C16:0	42.07 ^c^	48.08 ^b^	51.65 ^ab^	41.33 ^c^	52.67 ^a^	37.16	37.70	38.27	36.08	38.34	37.95
C16:1	2.24 ^b^	1.47 ^c^	3.35 ^a^	1.51 ^c^	3.22 ^a^	3.76 ^b^	4.07 ^ab^	4.36 ^a^	3.86 ^b^	3.91 ^b^	3.89 ^b^
C18:0	11.59 ^c^	13.86 ^a^	13.73 ^a^	12.37 ^bc^	12.93 ^ab^	9.90 ^b^	9.96 ^b^	10.36 ^b^	10.02 ^b^	11.22 ^a^	10.72 ^ab^
C18:1n9	19.15 ^a^	16.66 ^b^	10.84 ^d^	15.24 ^c^	7.87 ^e^	18.17	18.40	18.63	17.68	18.66	18.95
C18:2n6	15.43 ^b^	9.85 ^c^	3.92 ^d^	18.70 ^a^	3.63 ^d^	13.82 ^a^	13.79 ^ab^	13.55 ^ab^	12.71 ^b^	13.01 ^ab^	13.64 ^ab^
C18:3n3	1.15	0.65		1.93		2.25 ^a^	2.14 ^a^	1.97 ^b^	1.85 ^b^	1.89 ^b^	2.15 ^a^
C18:3n6						0.23 ^c^	0.25 ^bc^	0.29 ^a^	0.26 ^b^	0.25 ^bc^	0.26 ^b^
C20:0				0.34		0.35 ^bc^	0.36 ^bc^	0.36 ^bc^	0.34 ^c^	0.44 ^a^	0.38 ^b^
C20:3n3						0.13 ^a^	0.11 ^ab^	0.09 ^bc^	0.08 ^c^	0.09 ^bc^	0.10 ^bc^
C20:5n3				0.51	0.95	1.00 ^a^	0.87 ^b^	0.71 ^de^	0.67 ^e^	0.76 ^cd^	0.79 ^c^
C22:1n9						0.13 ^ab^	0.14 ^ab^	0.13 ^ab^	0.11 ^b^	0.15 ^a^	0.12 ^ab^
C22:2						0.05 ^a^	0.04 ^ab^	0.03 ^b^	0.03 ^b^	0.04 ^ab^	0.04 ^ab^
C22:6n3				0.23	0.40	2.98 ^a^	2.63 ^b^	2.22 ^de^	2.12 ^e^	2.44 ^bc^	2.33 ^cd^
C24:0						0.05 ^ab^	0.05 ^ab^	0.05 ^ab^	0.04 ^b^	0.07 ^a^	0.05 ^ab^
SFA ^3^	61.64 ^c^	71.37 ^b^	81.79 ^a^	61.42 ^c^	83.37 ^a^	54.07	54.28	54.82	51.78	55.55	54.63
MUFA ^4^	21.39 ^a^	18.13 ^b^	14.29 ^c^	17.21 ^b^	11.65 ^d^	23.85	24.37	24.90	23.30	24.51	24.69
PUFA ^5^	16.97 ^b^	10.50 ^c^	3.92 ^d^	21.37 ^a^	4.98 ^d^	22.08 ^b^	21.35 ^bc^	20.28 ^c^	24.92 ^a^	19.94 ^c^	20.68 ^bc^

^1^ Values are means from triplicate groups of shrimps where the values in each row with different superscripts are significantly different (*p* < 0.05). ^2^ Dry matter basis. ^3^ Saturated fatty acid. ^4^ Monounsaturated fatty acid. ^5^ Polyunsaturated fatty acid. ^6^ Commercial feed containing 47.44% protein and 5.65% lipid, Korea.

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
