# Peer review of "Potential of Bacterial Strains Isolated from Coastal Water for Wastewater Treatment and as Aqua-Feed Additives"

_microorganisms, 2021, doi:10.3390/microorganisms9122441_

Round 1

Reviewer 1 Report

It is not clear the impact on the manuscript of phylogenetic trees represented in figure 1. If this information is methodological evidence but not discussed in the paper core, it may be summarized in the methods texts.

In a feed-additive experiment, the authors conclude that  all assayed strains were associated with the higher fish-growth performance obtained same effects in B.safensis, and B. albus, and the best results were obtained in treatment using P. mariniglutinosa and P. celer

However, this last conclusion needs to complement, confirming data about the isolation of assayed four strains at the end of the fish-growth performance experiments.

Natural microbial communities or probiotic single assayed strains respond o environmental stress as temperature shock (freeze-dried bacteria biomass, cited in Line 120), or even later in the ecological environment provided by the fish native intestinal microbiome, in different ways. For example, it may involve cell death, evolving to dormant bacteria and across various microbial activities levels. Experimental data about metabolic capabilities of assayed strains shown in the left column of Tables 2, and 3 are, in fact, exciting because they validate the metabolic capabilities of assayed strains in LAB culture (line 132). But still, it is no evidence that these activities are also being performed inside the fish host).

I suggest that strain recovery data be included in the description of the results. Koch principles provide a robust experimental approach to ensure that results of probiotic strains' performance are adequately assigned.

Author Response

Reviewer #1:

1) It is not clear the impact on the manuscript of phylogenetic trees represented in figure 1. If this information is methodological evidence but not discussed in the paper core, it may be summarized in the methods texts.

Ans.) We revised the manuscript as suggested by the reviewer. The revised part is highlighted in red color in the manuscript on 178-184.

2) In a feed-additive experiment, the authors conclude that all assayed strains were associated with the higher fish-growth performance obtained same effects in B.safensis, and B. albus, and the best results were obtained in treatment using P. mariniglutinosa and P. celer. However, this last conclusion needs to complement, confirming data about the isolation of assayed four strains at the end of the fish-growth performance experiments.

Ans.) This study aims to evaluate the effect on fish growth by independently culturing/harvesting the microorganisms and adding them to feed in the form of powder (cell death). What is mentioned is another field.

3) Natural microbial communities or probiotic single assayed strains respond o environmental stress as temperature shock (freeze-dried bacteria biomass, cited in Line 120), or even later in the ecological environment provided by the fish native intestinal microbiome, in different ways. For example, it may involve cell death, evolving to dormant bacteria and across various microbial activities levels. Experimental data about metabolic capabilities of assayed strains shown in the left column of Tables 2, and 3 are, in fact, exciting because they validate the metabolic capabilities of assayed strains in LAB culture (line 132). But still, it is no evidence that these activities are also being performed inside the fish host). I suggest that strain recovery data be included in the description of the results. Koch principles provide a robust experimental approach to ensure that results of probiotic strains' performance are adequately assigned.

Ans.) It has nothing to do with this study and this (etiology) is out of the subject of this study.

Koch’s postulates

  1. The microorganism must be found in abundance in all organisms suffering from the disease, but should not be found in healthy organisms.
  2. The microorganism must be isolated from a diseased organism and grown in pure culture.
  3. The cultured microorganism should cause disease when introduced into a healthy organism.
  4. The microorganism must be reisolated from the inoculated, diseased experimental host and identified as being identical to the original specific causative agent.

I do appreciate your review’s consideration.

Reviewer 2 Report

Microorganisms

Manuscript: MDPI-1471878

The authors studied the effectiveness in biological-wastewater treatment, aqua-feed-additive applications in terms of fish, and studied removal of total ammonia nitrogen, nitrite, nitrate, total nitrogen and total phosphorus for the biological water treatment on fish growth using the microorganisms’ freeze-dries biomass. Four strains of bacteria were used. The authors concluded and reported that fur bacterial strains can be effectively utilized for biological wastewater treatment processes and as aqua-feed. This manuscript may provide interesting information in the field. Suggest check the terms and use the right terms. The authors reported total phosphorus, total nitrogen (TN), total ammonia nitrogen (TAN), nitrate (NO3-), and nitrite (NO2-). Suggest change the term nitrate (NO3-), and report in nitrate-N (nitrate as nitrogen); and change the term nitrite (NO2-), and report nitrite-N (nitrite as nitrogen).

  1. Abstract: Change NO3- and NO2- to nitrate-N and nitrite-N.
  2. Introduction: Change nitrite (NO2-) to nitrite-N, and change nitrate (NO3-) to nitrate-N.
  3. 3, 3.1: Change nitrate (NO3-) to nitrate-N, change nitrite (NO2-) to nitrite-N, and change phosphorus (TP) to total phosphorus (TP).
  4. 1: Change Total ammonia nitrogen (NH4+ + NH3, TAN, Figure 2a) to Total ammonia nitrogen (TAN, ionized ammonia as nitrogen plus ionized ammonia as nitrogen, Figure 2a). Change NO2- to nitrite-N, and change NO3- to nitrate-N. Check the term nitrogen dioxide [N2O].
  5. 1: Change dissolved organic nitrogen (DIN) to dissolved organic nitrogen (DON).
  6. Page 7, Fig. 1a, 1b, 1c, Y-axis: Change to TAN (total ammonia nitrogen, mg/L), Nitrite-N (mg/L), Nitrate-N (mg/L).
  7. Page 11: Change nitrite (NO2-) to nitrite-N, and change ammonia (NH3) to ammonia.
  8. Page 12: Change nitrite (NO2-) to nitrite and change ammonia (NH3) to ammonia.
  9. Page 12, Conclusion: Change nitrate (NO3-) to nitrate-N, and change nitrite (NO2-) to nitrite-N.

Author Response

Reviewer #2:

1) Abstract: Change NO3- and NO2- to nitrate-N and nitrite-N.

Ans.) We revised the manuscript as suggested by the reviewer. The revised part is highlighted in red color in the manuscript on line 15.

2) Introduction: Change nitrite (NO2-) to nitrite-N, and change nitrate (NO3-) to nitrate-N.

Ans.) We revised the manuscript as suggested by the reviewer. The revised part is highlighted in red color in the manuscript on line 45-47, 66.

3) 3, 3.1: Change nitrate (NO3-) to nitrate-N, change nitrite (NO2-) to nitrite-N, and change phosphorus (TP) to total phosphorus (TP).

Ans.) We revised the manuscript as suggested by the reviewer. The revised part is highlighted in red color in the manuscript on line 192, 200, 225, 227.

4) 1: Change Total ammonia nitrogen (NH4+ + NH3, TAN, Figure 2a) to Total ammonia nitrogen (TAN, ionized ammonia as nitrogen plus ionized ammonia as nitrogen, Figure 2a). Change NO2- to nitrite-N, and change NO3- to nitrate-N. Check the term nitrogen dioxide [N2O].

Ans.) We revised the manuscript as suggested by the reviewer. The revised part is highlighted in red color in the manuscript on Fig. 2.

5) 1: Change dissolved organic nitrogen (DIN) to dissolved organic nitrogen (DON).

Ans.) We revised the manuscript as suggested by the reviewer. The revised part is highlighted in red color in the manuscript on 203.

6) Page 7, Fig. 1a, 1b, 1c, Y-axis: Change to TAN (total ammonia nitrogen, mg/L), Nitrite-N (mg/L), Nitrate-N (mg/L).

Ans.) We revised the manuscript as suggested by the reviewer. The revised part is highlighted in red color in the manuscript on Fig. 1.

7) Page 11: Change nitrite (NO2-) to nitrite-N, and change ammonia (NH3) to ammonia.

Ans.) We revised the manuscript as suggested by the reviewer. The revised part is highlighted in red color in the manuscript on 330-331.

8) Page 12: Change nitrite (NO2-) to nitrite and change ammonia (NH3) to ammonia.

Ans.) We revised the manuscript as suggested by the reviewer. The revised part is highlighted in red color in the manuscript on 347.

9) Page 12, Conclusion: Change nitrate (NO3-) to nitrate-N, and change nitrite (NO2-) to nitrite-N.

Ans.) We revised the manuscript as suggested by the reviewer. The revised part is highlighted in red color in the manuscript on 362.

I do appreciate your review’s consideration.

Reviewer 3 Report

The authors had chosen an interesting topic. Unfortunately, the manuscript lacked sufficient description of methods and additions are definitely required. Minor comments have been listed below.

Abstract
Add a sentence to explain the background for the study.
14 1 hour from

Keywords
Aqua-feed-additives -> Aqua feed additives

Intro
36 remove field
39 environmental-friendliness
42 100 mg.L-1 -> 100 mg L-1 Check this throughout the MS!
55 on abolone Remove italics.
62 aqua feed additive

M&M
78 Define LB
112-116. Explain more thoroughly the IC equipment, column details, analysis method
and its validation data with LODs and LOQs.
142 Remove "samples of"
156 Explain much more thoroughly the GC method for determination of
fatty acids. Remember that based on this someone else should be 
able to replicate the experiment. Add LODs and LOQs as well.

Results
272 remove "one of the components of breast milk and a precursor to 272
inflammation-inducer prostaglandin and leukotriene" Not relevant here.
295 To conclude,
307 water quality improvement
309 water quality
316 Toxicity of ammonia

Author Response

Reviewer #3:

Abstract

1) Add a sentence to explain the background for the study.

Ans.) We revised the manuscript as suggested by the reviewer. The revised part is highlighted in red color in the manuscript on 11-12.

2) 14 1 hour from

Ans.) The manuscript was edited by English editing company. They say ‘in’ is right, not ‘from’.

Keywords

3) Aqua-feed-additives -> Aqua feed additives

Ans.) The manuscript was edited by English editing company. They say no problem using both. I appreciate the reviewer’s consideration, but personally I hope that this title will be selected.

Intro

4) 36 remove field

Ans.) We revised the manuscript as suggested by the reviewer.

The revised part is highlighted in red color in the manuscript on 38.

5) 39 environmental-friendliness

Ans.) We revised the manuscript as suggested by the reviewer. The revised part is highlighted in red color in the manuscript on 41.

6) 42 100 mg.L-1 -> 100 mg L-1 Check this throughout the MS!

Ans.) We revised the manuscript as suggested by the reviewer. The revised part is highlighted in red color in the manuscript on 44.

7) 55 on abolone Remove italics.

Ans.) We revised the manuscript as suggested by the reviewer. The revised part is highlighted in red color in the manuscript on 57.

8) 62 aqua feed additive

Ans.) As described above, I appreciate the reviewer’s consideration, but personally I hope that this title will be selected.

M&M

9) 78 Define LB

Ans.) We revised the manuscript as suggested by the reviewer. The revised part is highlighted in red color in the manuscript on 83.

10) 112-116. Explain more thoroughly the IC equipment, column details, analysis method and its validation data with LODs and LOQs.

Ans.) We revised the manuscript as suggested by the reviewer. The revised part is highlighted in red color in the manuscript on 117-123.

11) 142 Remove "samples of"

Ans.) We revised the manuscript as suggested by the reviewer. The revised part is highlighted in red color in the manuscript on 149.

12) 156 Explain much more thoroughly the GC method for determination of fatty acids. Remember that based on this someone else should be able to replicate the experiment. Add LODs and LOQs as well.

Ans.) We revised the manuscript as suggested by the reviewer. The revised part is highlighted in red color in the manuscript on 163-170.

Results

13) 272 remove "one of the components of breast milk and a precursor to 272 inflammation-inducer prostaglandin and leukotriene" Not relevant here.

Ans.) This is a further explanation of arachidonic acids for the reader. I appreciate the reviewer’s consideration, but personally I hope that this title will be selected.

14) 295 To conclude,

Ans.) I appreciate the reviewer’s consideration, but personally I hope that this title will be selected.

15) 307 water quality improvement

Ans.) The manuscript was edited by English editing company. They say no problem. I appreciate the reviewer’s consideration, but personally I hope that this title will be selected.

16) 309 water quality

Ans.) The manuscript was edited by English editing company. They say no problem. I appreciate the reviewer’s consideration, but personally I hope that this title will be selected.

17) 316 Toxicity of ammonia

Ans.) The manuscript was edited by English editing company. They say no problem. I appreciate the reviewer’s consideration, but personally I hope that this title will be selected.

I do appreciate your review’s consideration.

Round 2

Reviewer 3 Report

The authors have edited the MS and made some changes based on reviewers´ comments. However, there still are major issues lacking for example in methods describing the analyses used and still method validation data (or  its references) and LODs are missing. I cannot recommend publishing before these issues have been properly added.

Author Response

The authors have edited the MS and made some changes based on reviewers´ comments. However, there still are major issues lacking for example in methods describing the analyses used and still method validation data (or its references) and LODs are missing. I cannot recommend publishing before these issues have been properly added.

Ans.) We revised the manuscript as suggested by the reviewer. The revised part is highlighted in red color in the manuscript on 117, 125, 164, 171.

I do appreciate your review’s consideration.